# The Effect of the Laser Incidence Angle in the Surface of L-PBF Processed Parts

**Sara Sendino** [1],*, **Marc Gardon** [2], **Fernando Lartategui** [3], **Silvia Martinez** [1] and **Aitzol Lamikiz** [1]

1 Aerospace Advance Manufacturing Research Center-CFAA, P. Tecnológico de Bizkaia 202, 48170 Zamudio, Spain; silvia.martinez@ehu.eus (S.M.); aitzol.lamikiz@ehu.eus (A.L.)
2 EMEA AM Applications Manager, Renishaw Ibérica, Carrer de la Recerca 7, 08850 Barcelona, Spain; Marc.Gardon@renishaw.com
3 UO Structures & Statics, ITP Aero, P. Tecnológico de Bizkaia 300, 48170 Zamudio, Spain; fernando.lartategui@itpaero.com
* Correspondence: sara.sendino@ehu.eus

**Abstract:** The manufacture of multiple parts on the same platform is a common procedure in the Laser Powder Bed Fusion (L-PBF) process. The main advantage is that the entire working volume of the machine is used and a greater number of parts are obtained, thus reducing inert gas volume, raw powder consumption, and manufacturing time. However, one of the main disadvantages of this method is the possible differences in quality and surface finish of the different parts manufactured on the same platform depending on their orientation and location, even if they are manufactured with the same process parameters and raw powder material. Throughout this study, these surface quality differences were studied, focusing on the variation of the surface roughness with the angle of incidence of the laser with respect to the platform. First, a characterization test was carried out to understand the behavior of the laser in the different areas of the platform. Then, the surface roughness, microstructure, and minimum thickness of vertical walls were analyzed in the different areas of the platform. These results were related to the angle of incidence of the laser. As it was observed, the laser is completely perpendicular only in the center of the platform, whilst at the border of the platform, due to the incidence angle, it melts an elliptical area, which affects the roughness and thickness of the manufactured part. The roughness increases from values of $Sa$ = 5.489 μm in the central part of the platform to 27.473 μm at the outer borders while the thickness of the manufactured thin walls increases around 40 μm.

**Keywords:** roughness; incidence angle; additive manufacturing; L-PBF; INCONEL718

## 1. Introduction

Laser technology has been widely used in different industrial processes for many years, due to the characteristics of the laser source: high energy density, high efficiency, large temperature gradients, high repeatability of the process, and formation of a narrow heat-affected zone (HAZ) [1]. Nowadays, it is possible to find various research works on laser technologies such as laser welding [2,3], cutting [4,5], remelting [6,7], additive technologies [8,9] . . . thus demonstrating the importance of the development of these technologies.

Due to the great development that the different technologies based on metal additive manufacturing (AM) have had in recent years, it seems reasonable that one of the most researched and promising metal AM process is the Laser Powder Bed Fusion (L-PBF) technology. This technology, also known by commercial names as Selective Laser Melting (SLM) or Direct Metal Laser Sintering (DMLS), is being applied for the direct manufacturing of functional components in different sectors [10,11].

In the L-PBF process, the final part is manufactured directly using powder as raw material. This powder is homogeneously distributed by a recoater in constant thickness layers of 20 to 60 μm. Once the recoater has distributed a layer of powder, a laser beam melts the powder within an area, which has been previously specified in the CAD (Computer-Aided Design) file. Once the complete layer has been processed, the platform lowers the thickness of a layer and the process is repeated layer-by-layer until the full part is obtained [12].

This technology presents many advantages compared to conventional machining technologies such as the ability to manufacture geometrically complex parts [10,13,14]. In addition, L-PBF also presents advantages above other AM technologies as Spierings [15] showed. One of these advantages is the wide range of alloys with very different properties and characteristics that can be processed. Thus, different applications and studies can be found with high toughness Ti-TiB composites [16,17], high-density alloys such as Cu-10Sn bronze [18], high-resistant alloys such as $Al_{85}Nd_8Ni_5Co_2$ [19], or high modulus of elasticity and mechanical strength $CNTs/AlSi_{10}Mg$ [20]. Therefore, L-PBF process is of special interest in sectors such as health and aerospace due to the possibility of obtaining complex parts in a wide range of available materials [10].

However, L-PBF technology has several limitations, particularly related to the poor surface finish and the need for final post-processing of the manufactured parts [10]. The parts manufactured using L-PBF show roughness values Ra between 3.2 and 12.5 μm [13], whereas conventional technologies obtain roughness values between 1–2 μm [11].

Surface quality is a critical attribute in the aerospace and medical sectors since surface imperfections caused by roughness could impact the performance of the part. Therefore, surface roughness is a well-researched crack initiator [12], reducing the life-span of the end part due to lower tensile and fatigue strength [10]. In the medical domain, rough surfaces can accumulate more micro-organisms than a smooth surface, requiring a finishing operation in many applications in the health sector [21]. Furthermore, roughness has a significant effect on the tribological behavior of the surface in applications where the friction between surfaces is present.

Nowadays, in order to minimize the roughness and to achieve functional parts with real applicability, finishing operations are carried out. However, the finishing operation of parts manufactured using L-PBF technology is frequently very complex due to the geometrical complexity of the parts [22]. In addition, the finishing process increases considerably the manufacturing time and cost of the part [12] since most of the finishing processes involve manual operations. Moreover, sometimes it is not possible to finish some areas such as internal ducts or hollow cavities [23]. Therefore, it is necessary to reduce the roughness as much as possible in the additive manufacturing process itself before finishing the final part.

The resultant roughness can be originated by different factors. On the one hand, the process parameters during manufacturing play an important role in the resulting roughness [22,24–26]. In this sense, it is necessary to adjust parameters such as laser power, exposure time, the distance between points, and overlapping. Some previous studies state that the use of higher laser power reduces roughness, since it increases the wettability of each layer [12], while other studies show that excessive laser power can lead to spatters of excessively oxidized particles, which would also increase roughness [23]. In addition, excessive laser power will lead to higher porosity. Thus, it is necessary to find a balance between the laser power and scan speed to ensure acceptable roughness and other properties such as porosity [11,12,23]. In addition, another process parameter that influences the roughness is the thickness of the layer used [27].

In any case, even when the optimal parameters are being used, the surface roughness can be very different depending on the position of the part in the machine [28], due to the direction of the inert gas in the manufacturing chamber [29], or the angle of incidence [28] and the geometry being processed. One of the most affected areas by roughness is the downskin area. Downskin areas are those that have not had molten material in the lower layer and have been built over non-melted powder. In this case,

some of these powder particles remain partially attached to the processed layer when the layer melts and re-solidifies during manufacturing [30].

There may be other effects such as partial melting of powder particles due to heat transfer when several parts are manufactured on the same platform. When different parts are manufactured very close to each other, the accumulated heat of one part may affect the adjacent, causing its surface temperature to rise, resulting in partially melt powder particles that will attach to the surface [21]. A similar effect can be observed in the case of large parts, where a high number of layers need to be processed and the generated heat into the manufacturing chamber is also higher. It should also be noted that this roughness due to the particles attached to the surface will depend on the characteristics of the powder used [31,32].

Finally, a very relevant factor is the angle of incidence of the laser with respect to the powder layer. Since the laser beam is radiated from the top part of the machine and depending on the location of the processed area, the laser beam is focused on the powder bed at a certain incidence angle depending on the area where the part is being manufactured.

The influence of this effect has not been so analyzed by the literature in comparison with other effects such as powder distribution or process parameters. The study carried out by Kleszczynski et al. [28] analyses the impact of increasing the radial distance with respect to the center of the platform, due to the laser beam incidence angle. Since the laser beam is not completely perpendicular to the platform as can be seen in Figure 1 (figure adapted from [33,34]), an increase in the roughness of the parts is obtained.

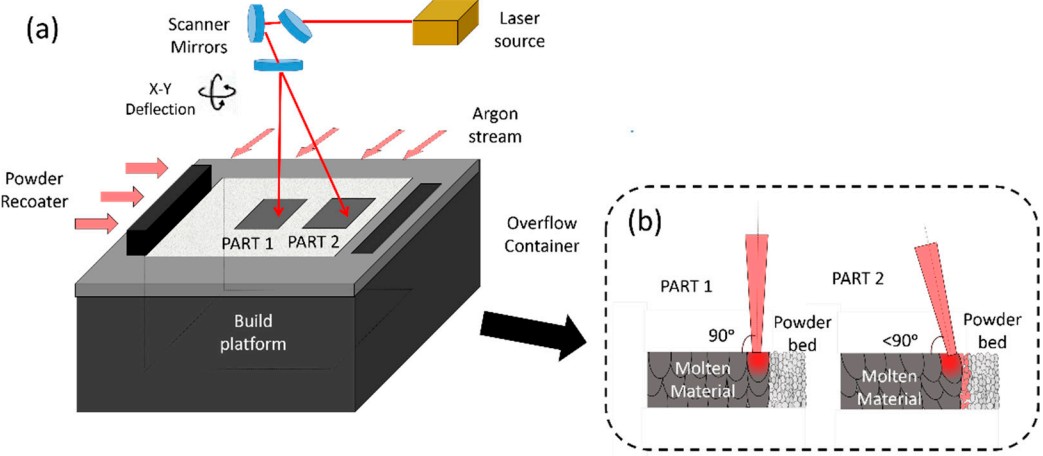

**Figure 1.** Effect of the incidence angle of the laser: (**a**) Diagram of the Laser Powder Bed Fusion (L-PBF) process showing the scanner operation and the possible angle of incidence in the different areas of the platform; (**b**) Detail of part 1 located in the central area of the platform and part 2 located at one side of the platform.

Throughout this article, the effect of the angle of incidence of the laser beam on the surface roughness is detailed, since, after an exhaustive bibliographic study, it seems to be a relevant but not sufficiently studied factor to minimize the resulting surface roughness obtained in the L-PBF process.

## 2. Materials and Methods

Two different tests were designed to analyze the incidence angle and check its effect on the final parts manufactured using L-PBF technology. All tests were carried out using the Renishaw AM400 manufacturing system (Renishaw, Stone, UK), which has a 400 W CW-M and a 3D scanner for laser beam movements with a maximum scanning speed of 7000 mm/s. This gives a reduced laser beam diameter of 70 μm in the focal point or working plane. The software used for programming the

parameters of the manufacturing process was QuantAM (version V4). The tests designed for this purpose are summarized below:

## 2.1. First Tests: Study of the Morphology of the Laser on the Platform

This study analyzed the effect of laser radiation on the different areas of the manufacturing platform. To carry out this test, the laser beam was radiated directly to the platform, without pre-depositing a powder layer. A grid of 25 × 25 uniformly distributed points was designed in order to test the entire workspace of the platform. The parameters were adjusted to analyze the points marked on the platform separately without any overlapping, as shown in Figure 2. In addition, the correct balance between the power and exposure time parameters was set to optimize the mark on the platform. Specifically, a power of 200 W, an exposure time of 200 µs, and a distance between points of 900 µm were selected.

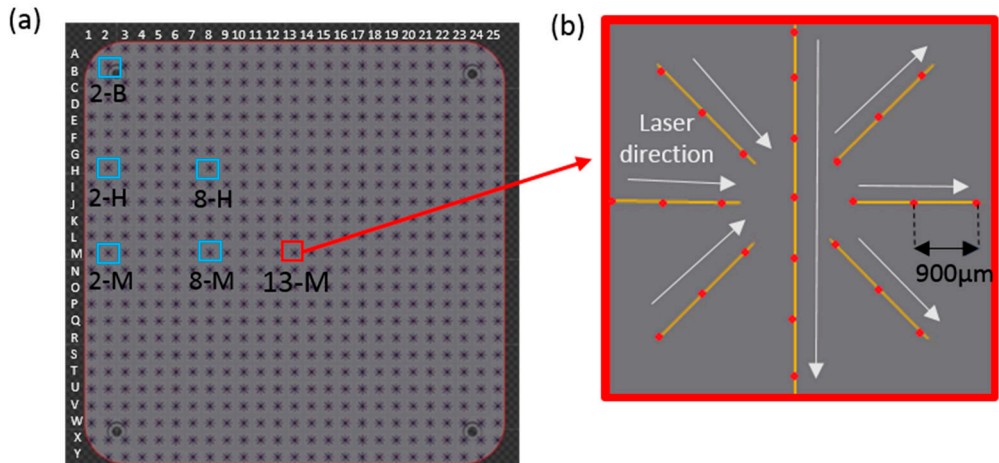

**Figure 2.** Test design to analyze the influence of laser inclination angle on the platform: (**a**) Distribution of the marks in the platform.; (**b**) Marks made on the platform in a star shape defining the laser strategy used.

Once the points were marked in the different areas of the platform, the morphology and dimensions of these points were studied using an infinite focus microscopy (Infinite Focus Microscope model Control Server FP G1 Vf2, Alicona, Raaba/Graz, Austria), using 500× magnification.

The different marks were made in a star shape and distributed evenly throughout the platform. With this star shape, it was intended to change the direction of the laser in order to be able to analyze the distortion caused by the angle of incidence independently of this direction of the laser and the possible delays related to the scanner.

From each of the stars marked on the platform, 10 of the points were analyzed and measured by the software Alicona MeasureSuit of the Infinite Focus Microscope measurement system. Thanks to this system, it was possible to estimate two radiuses from each of the elliptical or circular points, and by dividing these two radiuses, the aspect ratio of the ellipse could be calculated, as can be seen in Equation (1). In the case of circular marks, the value of this ratio was equal to one.

$$\text{Ellipse aspect ratio} = \frac{\text{Ellipse major axis}}{\text{Ellipse minor axis}} \tag{1}$$

Thanks to this study, it was possible to analyze the circularity of the marks made by the laser on the different parts of the platform.

## 2.2. Second Test: Effect of the Angle of Incidence of the Laser on Parts Manufactured by L-PBF Technology

Inconel 718 powder specially designed for this technology was used to manufacture these test parts. The powder shows high sphericity (over 80% of the particles showed 100% sphericity) and a particle size between 15–45 µm. Because of these specifications, the powder can flow and distribute correctly in each layer. In addition, powder meets the specific chemical composition of Inconel 718, shown in Table 1.

**Table 1.** Chemical composition in weight percentage of Inconel 718.

| Elements | Ni | Cr | Co | C | Mo | Al | Ti | Fe | Nb | Si and Mn | P and S | Cu | B |
|---|---|---|---|---|---|---|---|---|---|---|---|---|---|
| %Weight | 50–55 | 17–22 | ≤1 | ≤0.08 | 2.8–3.3 | 0.2–0.8 | 0.65–1.15 | Bal | 4.75–5.5 | ≤0.35 | ≤0.015 | ≤0.3 | ≤0.006 |

Three different geometries were designed and analyzed in order to study three different effects: Surface roughness, internal microstructure, and thin walls thickness.

### 2.2.1. Test Part A: Roughness Measurement (Red Parts)

These parts were designed specifically for measuring surface roughness, using an infinite focus 3D measurement system with 500× magnification, according to ISO 25178 [35].

The geometry designed for this purpose had a study surface of 13 mm × 13 mm, where three measurements of 1 mm × 13 mm on each of these surfaces were made, always analyzing the outside surface of each part as can be seen in Figure 3. The arithmetical mean height of the surface (*Sa*) and the maximum height of the surface (*Sz*) values were obtained and the standard deviation of these measurements was calculated to determine the error bars. In addition, thanks to this optical measurement system, surface topographies could be obtained and thus the particles partially adhered to the surface could be also analyzed.

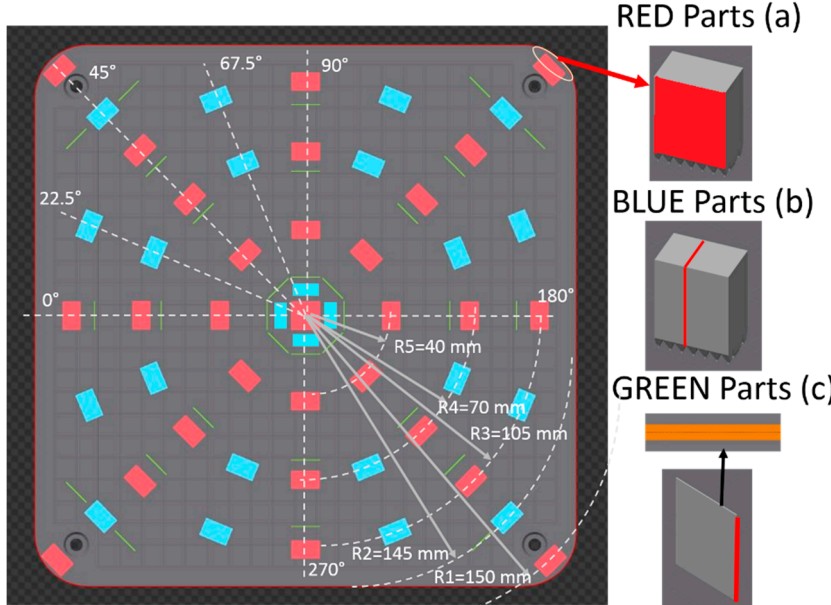

**Figure 3.** Design of the second platform: (**a**) parts for the roughness measurement; (**b**) parts for the microstructure analysis; (**c**) parts for measuring the thickness of thin walls.

The process parameters were the usual set for the manufacture of Inconel 718 parts using L-PBF technology using a layer thickness of 60 µm. Specifically, the volume parameters were a laser power of 200 W, exposure time of 70 µs, a distance point and hatch distance of 80 µm, a strategy angle variation of 67°, and the border parameters were a laser power of 125 W, exposure time of 75 µs, a point distance

of 20 µm, and a layer thickness of 60 µm. These parts were distributed on a platform with a radial distance of 0, 40, 75, 105, and 150 mm from the center.

### 2.2.2. Test Part B: Microstructure Analysis (Blue Parts)

These parts were specially designed to analyze their internal microstructure. The geometry used was the same as for Parts A but the parameters were changed in order to study the desired effect. Specifically, these parts were manufactured without borders and using a 0° variation of the strategy angle so that the Gaussian-shape melted areas were aligned in order to analyze them. These parts were distributed on a platform with a radial distance of 20, 75, 105, and 145 mm from the center.

To analyze these parts, a section was cut and encapsulated. To analyze always the same surface, all parts were cut in the same direction (as shown in Figure 3b) and these samples were encapsulated using phenolic resin.

Then, the planar grinding step was made using Silicon carbide and corundum sandpaper of lower grain size of FEPA (Federation of European Producers of Abrasives) 400, to achieve a surface ready to be polished, four steps were made (using FEPA 400, FEPA 600, FEPA 800, and FEPA 1200 grain size). Once the surface was ready, it was polished using a diamond polycrystalline of 1 and 3 µm.

Once the mirror-finished surface was achieved, this surface was chemically attacked using the 25 Marble etchant using the swabbing method, as specified in the ASTM E 407 standard [36] and with a composition of 4 g $CuSO_4$, 20 cc HCl, and 20 cc $H_2O$. Finally, surface images were taken using the infinite focus 3D measurement system with 200× magnification.

### 2.2.3. Test Part C: Thin Walls Thickness Measurements (Green Parts)

Finally, thin walls of 13 mm height were designed and placed in different areas of the platform to measure their thickness. These test parts were built with a single laser track, and the same border parameters used in the "A Test Parts".

Once these parts were manufactured, the thickness of each test part was measured using the infinite focus 3D measurement system with 200× magnification. Figure 4 shows the methodology used for the analysis of thin walls. The Image J image analysis software (version ImageJ 1.x [37]) was used for extracting the contour of the images of the infinite focus microscope. This contour was then approximated to ellipses to define the thickness of the part, which would be equivalent to the average value of the vertical radius of the ellipses.

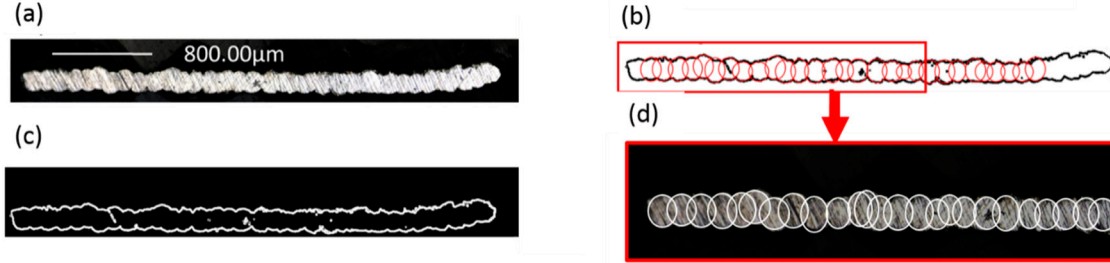

**Figure 4.** The methodology used for the analysis of thin walls: (**a**) image obtained through the infinite focus microscope; (**b**) extraction of the thin wall contour; (**c**) approximation of the ellipses to the generated contour; (**d**) checking this approximation in the obtained image.

Twenty-five measurements were made on each of the thin walls and the average value and standard deviation between them were calculated; therefore, and taking into account that eight thin walls were manufactured for each radial distance, 200 measurements were obtained in each of the radial distances (these radial distances being 20, 70, 100, and 145 mm).

## 3. Results

### 3.1. First Tests: Study of the Morphology of the Laser on the Platform

Analyzing the laser marks on the platform surface, it could be seen that these marks in the central part of the platform present a circular shape while marks located far from the center acquire an elliptical appearance. In addition, all the ellipses showed its largest radius in the radial direction towards the center of the platform.

Figure 5a shows this effect on the images obtained when analyzing the center point of stars with different positioning on the platform. As can be seen, the central area of the platform shows circular marks, with a diameter of around 200 μm, while in the case of the marks located away from the centre, elliptical marks up to 240 μm are observed. Therefore, a considerable difference between the different laser marks on the platform is observed depending on the location of the laser mark.

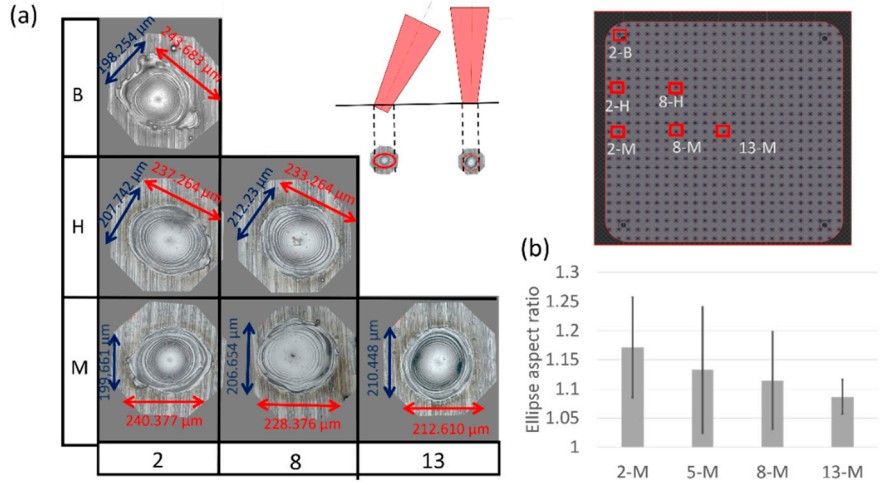

**Figure 5.** Analysis of the marks made by the laser on the platform: (**a**) diagram of some of the images obtained when at the central mark of the stars was analyzed (2-B, 2-H, 2-M, 8-M, 8-H, and 3-M); (**b**) ellipse aspect ratio of different stars analyzed.

In addition, the ellipse aspect ratio was calculated. It is observed in Figure 5b that this ratio was closer to unity in the tests near the center of the platform, while in the marks located on the sides of the platform, this ratio increase, and the circularity of the marks decrease.

### 3.2. Second Test: Effect of the Angle of Incidence on Parts Manufactured by L-PBF Technology

#### 3.2.1. Test Part A: Roughness Measurement

The results obtained with the roughness measurement are shown in Figure 6. Despite manufacturing the same part with the same parameters and the same raw powder, surface roughness is influenced by the position of the part on the platform. As it can be observed, higher surface roughness is measured for parts located away from the center of the platform. Results show that the roughness is dependent on the position of the part on the platform because of the angle of incidence of the laser.

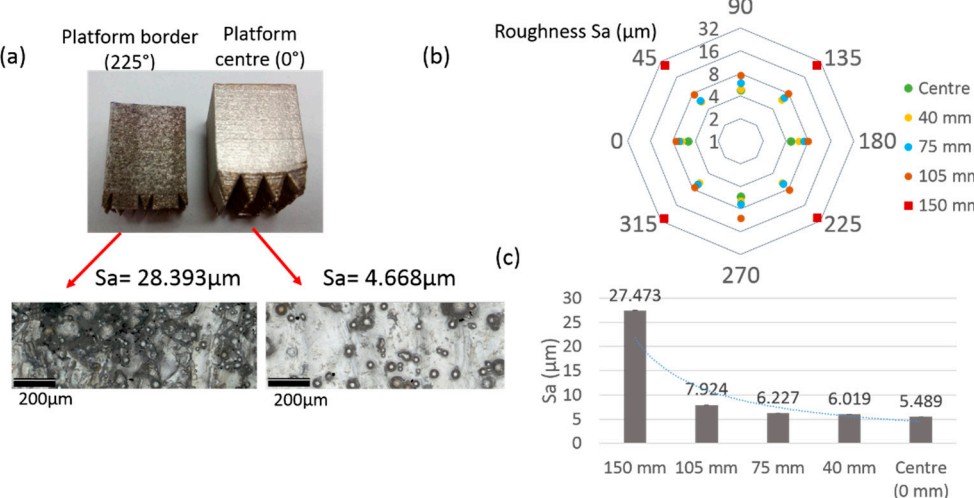

**Figure 6.** Roughness results obtained: (**a**) image and topographies of maximum and minimum roughness values: maximum roughness (225° and radial distance of 150 mm) and minimum roughness (center 0°); (**b**) mean value of the roughness for different radial distance from the center of the platform; (**c**) roughness mean value results obtained each test part.

As it is observed in Figure 6, the relatively high roughness and poor surface finish of the parts located at the borders of the platform can be seen with the naked eye. Analyzing the surface topographies, it could see how in the parts manufactured in the central part of the platform the layers of the part and some particles partially adhered to the surface can be differentiated; however, in the parts located in the outer border of the platform, neither the layers nor the particles partially adhered to the surface can be differentiated so clearly, but the surface quality has considerably worsened due to the melting of the powder.

It has also been determined that the roughness does not increase linearly, but increases significantly at the borders of the platform while the roughness remains at similar values in the center.

### 3.2.2. Test Part B: Microstructure Analysis

The microstructure has been also studied to analyze the possible influence of the incident angle on part integrity. Figure 7 shows the microstructure of two sections corresponding to the central area and the peripheral border of the platform. While the section corresponding to the part in the central area shows a uniform pattern of laser tracks, with the melted areas oriented in a vertical direction, the part in the border shows a less uniform pattern with the melted areas distorted due to the effect of the laser orientation. This effect is shown schematically in Figure 7.

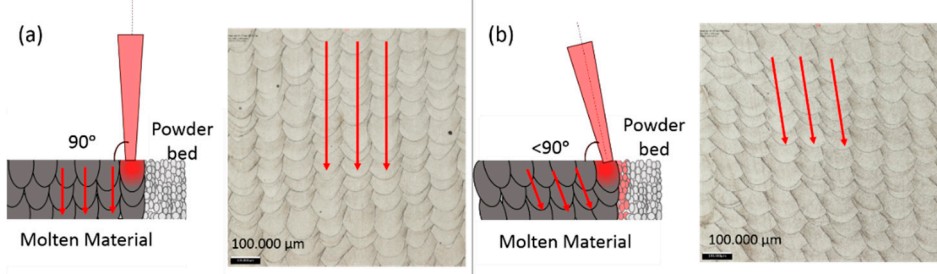

**Figure 7.** Analysis of the microstructure in the different parts located at 45°: (**a**) central part (radial distance from the center 20 mm); (**b**) Part located at the edge of the platform (radial distance from the center 145 mm).

Nevertheless, although the laser pattern creates a different microstructure due to the angle of the laser beam, it is observed that both components are free of porosity and defects and the mechanical properties of both specimens meet the required requirements.

### 3.2.3. Test Part C: Thin Walls Thickness Measurements

To conclude this study, thin walls corresponding to Test Part C and manufactured using single laser tracks were analyzed. The results of these tests are shown in Figure 8. Figure 8a shows the effect of the angle of incidence on the thickness for the different test parts manufactured in one of the diagonals of the platform (at 45°, according to Figure 3).

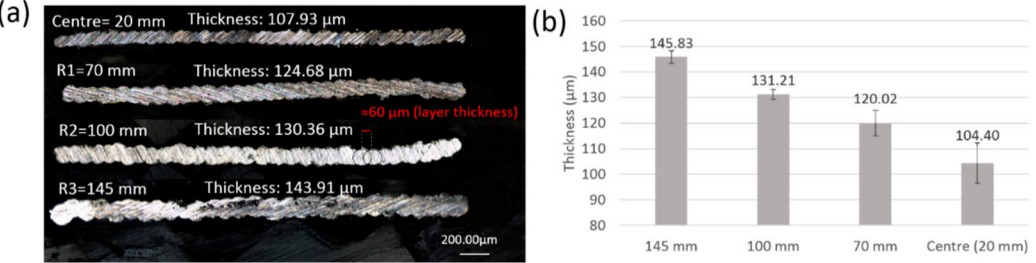

**Figure 8.** Analysis of the thin walls manufactured at 45°: (**a**) Measurements of wall thickness for test parts manufactured at 45° and a different distance from the platform center; (**b**) Mean value of the thickness for different radial distances from the center of the platform.

The influence of the radial distance from the center of the platform in the thickness is observed in Figure 8b) where the results show a gradual increase of the thickness as the radial distance increases. In particular, the thickness of these walls increased more than 40 μm in the parts located at the borders of the platform. Moreover, this result is consistent with the analysis of the previous tests.

## 4. Discussion and Conclusions

On the one hand, the measurement of the geometry of the laser spot at the platform surface demonstrated that this spot is not circular at the borders of the platform and takes an elliptical shape, which causes position-dependent laser radiation and a distortion in the melt pool. This effect is due to the technology itself, the lens used in the scanner, and the physics of the process. In this technology, and in general, in all laser technologies that require a flat working field, F-theta lenses are used, which ensure that all the main beams on the image side are parallel to the optical axis, resulting in the perpendicularity of the beam with respect to the platform.

On the other hand, results show that this angle of incidence influence the surface quality, microstructure, and thickness of the manufactured parts. As a consequence of the elliptical shape of the laser spot on the areas further away from the center, the area of powder bed heated is different from the central area and the melting of this powder does not occur in the same way. This effect is observed in Test Part A.

Furthermore, after the analysis of Test Part C corresponding to the thin walls, the effect of the angle of incidence could also be seen, as the distortion of the laser spot has resulted in a larger melted area, which has increased the thickness of the walls of the parts located in the outer borders of the platform.

Finally, after this exhaustive study, it has been possible to see how this angle of incidence has a relevant effect on the roughness of the part (Test Part B).

It should also be noted that this effect can have a considerable effect, as the increase in the diameter of the laser spot on the platform can reach up to 40 μm and different effects can occur in the manufactured part:

- Increase of the thickness of the thin walls: These walls were manufactured with a single laser track and the thickness increase was very similar to the laser spot size enlargement at the borders of the platform.
- Roughness increase: Similarly, the results of the roughness measurements are also coherent with the results of the other tests. Thus, an increase in roughness is observed as the test parts are manufactured away from the center of the platform. The high roughness, in this case, is due to the poor surface quality caused by the non-circular shape of the melt pool.
- Regarding microstructure, although a different pattern is observed due to the inclination angle and the distortion of the melt pool, similar mechanical properties are obtained, and no porosity or cracks have been observed.

Due to the great effect of the angle of incidence on the different parts manufactured and on the surface quality of the parts, it has been decided to continue with this study in future investigations. For this purpose, it is expected to characterize the laser spot on the platform, in order to determine this angle of incidence with high precision. Next, based on the shape and characteristics of the spot, the melt pool generated will be analyzed to try to obtain a direct relationship between the angle of incidence and the roughness of the parts.

**Author Contributions:** Conceptualization, S.S., S.M., and A.L.; methodology, S.S., S.M., M.G. and F.L.; experimental test S.S. and S.M.; result analysis, S.S., S.M., M.G., F.L. and A.L.; resources, S.M. and A.L.; writing—original draft preparation, S.S. and S.M.; writing—review and editing, S.S., S.M. and A.L.; supervision, M.G., F.L. and A.L.; project administration, M.G., F.L. and A.L.; funding acquisition, A.L. All authors have read and agreed to the published version of the manuscript.

**Funding:** This research was funded by the Spanish Ministry of Industry and Competitiveness under the CDTI JANO project and by Basque Government (Eusko Jaurlaritza) under the ELKARTEK Program, QUALYFAM project, Grant No. KK-2020/00042.

**Conflicts of Interest:** The authors declare no conflict of interest.

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
