# Peer review of "The Effect of the Laser Incidence Angle in the Surface of L-PBF Processed Parts"

_coatings, doi:10.3390/coatings10111024_

Round 1

Reviewer 1 Report

This work by Sendino et al. investigates the effect of the laser incidence angle on the L-PBF processed surface. In my opinion, this work is well-designed and well-written. I think it can be accepted after a minor revision.

Some typos mus be corrected:

lines 175-176. Please, use subscript in chemical formulas

line 231. "otter boder" must be corrected

line 241. "beorder"

line 271 these areas instead of "this areas"

Author Response

Thank you for reviewing this manuscript and indicating the corrections and improvement proposals. The authors honestly believe that the amendments have been very helpful for improving the quality of the document. We have corrected all the typos that have been pointed out:

  • lines 175-176. Please, use subscript in chemical formulas: (Specified in the ASTM E 407 standard and with a composition of: 4g CuSO4, 20cc HCl and 20 cc H2O)
  • line 231. "otter boder" must be corrected However, in the parts located in the outer border of the platform
  • line 241. "beorder" the part in the border shows a less uniform pattern with the melted areas distorted due to the effect of the laser orientation
  • line 271 these areas instead of "this areas" As a result of this elliptical shape, the area of powder bed heated at these areas is different to the central

Reviewer 2 Report

Good work has been done by the authors on surface roughness based on the angle of incidence of the laser beam. The manuscript is recommended for publication. 

Author Response

Thank you for reviewing the work and your assessment. As reviewer suggest authors have reviewed the document to try to correct the typos throughout the document. Such as line 231. "otter boder"→ outer border.

Reviewer 3 Report

Dear Authors,

The reviewed submission is an experimental work and describes the results of research on modern laser manufacturing techniques included in the group of additive manufacturing processes. The subject matter is one of the most up-to-date and currently developed research trends in manufacturing engineering and material science. I believe that the article should be published after introducing some important changes and detailed text correction.

Title: I propose to shorten the title: "The effect of the laser incidence angle in the surface of L-PBF processed parts". In my opinion, this does not result in a loss of information.

Abstract: Lines 11/12: the abbreviation should be explained. Please provide quantitative results of your work at the end of the abstract.

Keywords: please add the keyword of the research material (this is important from the point of view of the title and profile of the journal).

Figures: please describe the figures in accordance with the journal's guidelines: (a), (b) ...

The introduction section is well prepared from the point of view of additive manufacturing. Please consider extending the paragraph with a broader context on the potential use of the laser beam in manufacturing techniques. In this regard, I can recommend the current MDPI literature: https://www.mdpi.com/1996-1944/13/11/2653, https://www.mdpi.com/1996-1944/13/13/2930, https://doi.org/10.3390/ met10101294. This will also allow you to expand the References section.

Figure 1: Please consistently write the second words of the description, either in upper or lower case.

Please add the laser power (400 W?).

Chapter 2:

Please specify the chemical composition of Inconel 718 powder.

Lines 146 and 147: The units "mm2" is not correct: these are linear dimensions.

Chapter 3: Figure 6 (b): font is too small.

Figure 8: change commas to dots.

Chapter 4: The commentary resulting from the comparison of the presented results with information from the literature is clearly missing in this part of the article. There is a lot of works in the literature on SLM of nickel alloys.

References: in my opinion, an article on such a modern subject should have at least 25 current literature items (articles from the last 2 years). The bibliographic description is not compliant with the journal's guidelines. Please pay attention to the exact writing of the names, eg in [14] Kleszczynski. Please correct the typos, e.g. line 341: dtuctures, Precedia.

Please cite the references in a manner consistent with the journal's guidelines.

Author Response

Thank you for your comments, I attach below the point by point response:

Reviewer 4 Report

The study looks good, but please consider the following suggestions prior to acceptance;

  • In literature review, try to add more recent studies from 2019- 2020. 
  • How the performance of different samples can be linked with laser incidence angle? May be you can plan some testing of samples to provide this detail.
  • Conclusions should be presented in a manner that relate the physics of the process as well.

Author Response

Thank you for your comments, I attach below the point by point response

Round 2

Reviewer 3 Report

Dear Authors, 

Thank you very much for taking into account my comments and for all the replies I consider warranted. I am convinced that your manuscript is very good and should be published.
Best regards,

Reviewer 4 Report

It looks good now.